# Polyphenols and Fibre: Key Players with Antioxidant Activity in Two Extracts from Pomegranate (*Punica granatum*)

**DOI:** 10.3390/ijms26199807

**Published:** 2025-10-09

**Authors:** Jessica Maiuolo, Federico Liuzzi, Francesca Oppedisano, Anna Spagnoletta, Rosamaria Caminiti, Valeria Mazza, Saverio Nucera, Salvatore Ragusa, Luigi Tucci, Giuseppe Trunfio, Lucia Carmela Passacatini, Sara Ilari, Giancarlo Statti, Vincenzo Mollace, Carolina Muscoli

**Affiliations:** 1Department of Health Science Institute of Research for Food Safety & Health (IRC-FSH), University “Magna Graecia” of Catanzaro, 88100 Catanzaro, Italy; maiuolo@unicz.it (J.M.); r.caminiti@unicz.it (R.C.); valeria.mazza@unicz.it (V.M.); l.tucci@head-sa.com (L.T.); g.trunfio@head-sa.com (G.T.); carmela.passacatini@sanraffaele.it (L.C.P.); sara.ilari@sanraffaele.it (S.I.); mollace@libero.it (V.M.); muscoli@unicz.it (C.M.); 2Laboratory for Techniques and Processes in Biorefineries, ENEA—Trisaia Research Centre, S.S. Jonica 106, Km 419+500, 75026 Rotondella, Italy; federico.liuzzi@enea.it; 3Laboratory “Regenerative Circular Bioeconomy”, ENEA—Trisaia Research Centre, S.S. Jonica 106, Km 419+500, 75026 Rotondella, Italy; anna.spagnoletta@enea.it; 4PLANTA/Research, Documentation and Training Center, 90121 Palermo, Italy; ragusa@unicz.it; 5Department of Pharmacy, Health and Nutritional Sciences, University of Calabria, 87036 Cosenza, Italy; g.statti@unical.it

**Keywords:** pomegranate fruit, *Punica granatum*, WFE and IME, antioxidant properties, dietary fibre

## Abstract

The pomegranate fruit offers numerous health benefits to humans due to its rich composition of various chemical components, including polyphenols, fibre, flavonoids, minerals, vitamins, organic acids, alkaloids, and amino acids, among others. The antioxidant properties of pomegranate are well known, and this study aims to compare these activities in two extracts obtained from the fruit (“Whole Fruit Extract”, WFE and “Internal Membranes Extract”, IME). Various experiments were conducted using both extracts: (1) quantification of polyphenols and flavonoids using the Folin–Ciocalteu colorimetric assay and the aluminium chloride assay, respectively; (2) the measurement of the antioxidant activity was carried out by Reducing Power, Chelating Activity of Ferrous Ions (Fe^2+^), Radical Absorbance Capacity of Oxygen, Free Radical Scavenging Activity DPPH, and antioxidant effect in vitro; (3) quantitative and quantitative evaluation of the fibre was performed. IME has demonstrated a significantly greater antioxidant effect than WFE, despite possessing a smaller amount of both polyphenols and flavonoids (polyphenols: 68 mg GAE/g for WFE; 47 mg GAE/g for IME; flavonoids: 51mg QE/g for WFE; 35 mg QE/g for IME). For this reason, we evaluated the fibre composition in both extracts. The higher amount of glucans, xylans, and pectin in IME suggested that these fibrous components may be responsible for the greater antioxidant effect detected compared to WFE.

## 1. Introduction

*Punica granatum* is a plant belonging to the family *Lythraceae*, and it is native to Iran and northern India. Still, for climatic reasons (the plant prefers a temperate–warm climate with mild winters and hot summers), it has spread throughout the world, particularly in all countries in the Mediterranean region [1]. Botanically, the fruit of the pomegranate is a berry similar to the apple; its size can change according to the variety of the species and the growing conditions, but when the fruit is ripe, the diameter is between 5 and 12 cm [2]. The outer portion of the berry is reddish-brown, depending on the degree of ripening; in addition, the peel appears smooth and leathery. The inner part has several partitions separated by a soft white membrane called “cica”; this membrane separates the grains (up to 600 or more per fruit), also known as arils, that contain the seeds. The seeds are surrounded by a translucent pulp that ranges from white to ruby red, varying in acidity, sweetness, and fragrance [3,4]. Therefore, this description allows for dividing the pomegranate into two portions: a non-edible portion, consisting of the peel and membranes cica, and an edible portion, represented by the pulp and juice. The non-edible part makes up about 45–50% of the whole fruit and is usually discarded as waste [5]. Pomegranate is widely used in phytotherapy due to its numerous beneficial properties for human health, as it has been shown to exert a protective metabolic effect, offering better control of diabetes, hyperlipidaemia, and hypertension. In addition, its antioxidant, anti-inflammatory, antibacterial, antiviral, and anticancer activities have been extensively documented [6]. The beneficial properties of pomegranate are possible thanks to several chemical components such as polyphenols, flavonoids, fibre, minerals, vitamins, organic acids, alkaloids, and amino acids, among others [7]. These chemical components are found in all parts of the fruit; however, it is essential to note that the non-edible parts of the pomegranate contain more phytochemicals than the pulp, highlighting them as a potential source of bioactive ingredients [8]. The amount of recovered phytochemicals depends closely on the solvent used for extraction and may therefore be different [9].

Dietary fibre is a component of plant-based foods that the human body is unable to digest or absorb completely in the small intestine [10]. It is a complex mixture of polysaccharides and oligosaccharides, such as cellulose, hemicellulose, pectin, β-glucans, hydrocolloids, fructo-oligosaccharides, gum, and lignin [11]. These compounds differ in functionality and physiological effects after consumption; furthermore, the composition of dietary fibre can vary with the source [12].

The digestibility of these compounds depends primarily on their molecular size; for example, small sugars, consisting of at most two monomers or oligosaccharides composed of 3–9 monomers, can be digested. Conversely, polysaccharides with more than 10 monomers are indigestible [13].

A second difference distinguishes the fibre based on its solubility in water. There is soluble fibre, which dissolves completely in water, forming a gelatinous substance that can help absorb sugars and reduce blood cholesterol [14]. The insoluble fibre increases the volume of stool, facilitates intestinal transit, and contributes to the health of this organ [15].

In the 1970s, the scientific community began to observe that the daily consumption of fibre, as part of an appropriate dietary pattern, was associated with the control of long-term weight, maintenance of metabolic function (reducing risk of type 2 diabetes, and metabolic syndrome), optimal composition of the gut microbiota, and a reduced risk of developing cardiovascular diseases [16]. To date, additional health benefits have been attributed to fibre intake, such as a reduced risk of developing malignant colorectal and lung cancers, maintenance of gastrointestinal health (against irritable bowel syndrome, inflammatory bowel diseases, constipation, and diverticular disease), a reduction in chronic inflammation, and a slowdown of the progression of neurodegenerative diseases [17,18,19]. To facilitate the beneficial effects of fibre, specific intake values are recommended in European countries and the United States: 30–35 g per day for men and 25–32 g per day for women [20]. However, the population does not meet the recommended daily fibre intake, with typical consumption being about half of the appropriate level. Fruits are a very fibre-rich food, possess a moderate energy density, and contain countless beneficial molecules, such as polyphenols, vitamins, amino acids, minerals, and sugars that work synergistically to support a wide range of health benefits [21,22].

Since the pomegranate fruit has a high antioxidant potential, this scientific article aimed to compare the antioxidant properties of two extracts: WFE and IME. It is important to remember that although there are numerous data on the protective effects of pomegranate extracts, there are very few results on the extract from the internal membranes of the fruit. In this manuscript, these two extracts are compared for the first time.

The portions from which the extracts were obtained are shown in Figure 1.

## 2. Results

### 2.1. Yield of Polyphenols and Flavonoids in the Extracts WFE and IME

First, it was necessary to determine which solvents to use for dissolving the extracts. To properly dissolve WFE and IME, we used several solvents, preferring the one that provided the greatest amount of polyphenols and flavonoids, as determined by the Folin–Ciocalteu colorimetric assay and the aluminium chloride assay, respectively. The solvents compared were water, methanol, ethanol, dimethyl sulfoxide, and a hydroalcoholic solution consisting of methanol-water (50:50). Our need was to dissolve plant extracts from which to obtain polyphenols and flavonoids. Since these compounds have complex structures and have both polar and non-polar characteristics, we decided to use polar (water, ethanol) and medium-polar (methanol, dimethyl sulfoxide) solvents. In particular, dimethyl sulfoxide and methanol can dissolve many polar and non-polar substances, thanks to their ability to form dipole–dipole interactions [23,24]. Non-polar solvents were not used, as this work was carried out in vitro, and their characteristics would have affected the absorption of polyphenols and flavonoids within an organism.

#### 2.1.1. WFE Dissolves Appropriately in DMSO

As can be seen in Figure 2, the content of both polyphenols and flavonoids is significantly higher when WFE is dissolved in DMSO than in other solvents. For this reason, WFE has always been dissolved in DMSO, as reported in the results of this paper.

#### 2.1.2. Dissolution of the Extract IME

We conducted the same experiment on the IME extract, but obtained a different result. In this case, DMSO was not particularly effective, and solvents appeared to have approximately the same extraction capacity. The solvent that allowed for a significantly higher yield of polyphenols and flavonoids was a hydroalcoholic solution of methanol and water (50:50), as reported in Figure 3. In light of these results, we continued our work by dissolving WFE in DMSO and IME in a hydroalcoholic solution of methanol and water (50:50). Control cells were exposed to vehicles used to dissolve WFE and IME; this did not affect the results. Thus, the control depicted in the figures consists of untreated cells.

### 2.2. WFE and IME Are Not Toxic When Administered to Human Neurons

Subsequently, it was considered necessary to choose the most suitable concentration of WFE and IME to be used. We tested the effects on cell viability of different concentrations of extracts on human SH-SY5Y neurons. As can be seen from the magnification of Figure 4a, reported in Figure 4b, the concentrations tested (5–400 μg/mL) were not harmful to the cells, highlighting values that could be superimposed on the untreated cells. For this reason, we arbitrarily chose the concentration of 100 μg/mL, an intermediate concentration, which was neither too low and potentially ineffective, nor too high and possibly toxic.

### 2.3. Antioxidant Properties of WFE and IME

All the tests conducted in this experimental paper have shown that both WFE and IME extracts exhibit high antioxidant properties. Specifically, the measurement of antioxidant activity was carried out by reducing the power assay, ferrous ions (Fe^2+^) chelating activity [25], radical absorbance capacity of oxygen [26], free radical scavenging activity DPPH [27], and antioxidant effect in vitro [28]. However, it is correct to point out that IME has a greater effect than WFE.

#### 2.3.1. Reducing Power and Ferrous Ion (Fe^2+^) Chelating Activity

Reducing power is a measure of a compound’s ability to donate electrons and neutralise harmful free radicals. In this way, a compound may donate electrons, reduce other molecules, and act as an antioxidant. We have measured the ability of WFE and IME to reduce ferric ions to ferrous ions. As reported in Figure 5a, it is possible to appreciate a gradual increase in the absorbance of extracts, and this effect indicates an increase in their reducing power. In addition, IME’s absorbance value is significantly higher than WFE’s at each concentration used, indicating its higher antioxidant activity.

The ferrous ion chelating assay evidences the ability of a substance to bind to ferrous ions (Fe^2+^), inhibiting their participation in reactions that may lead to oxidative damage: the results obtained are represented in Figure 5b, and show that both extracts promote chelation of Fe^2+^ ions, helping to prevent the formation of harmful radicals. Again, IME showed significantly greater activity than WFE.

#### 2.3.2. Oxygen Radical Absorbance Capacity (ORAC)

The ORAC assay was performed to evaluate the scavenging properties of extracts WFE and IME against radical species, and the results are shown in Figure 6, panel a. As can be appreciated, their curves are between Trolox at a concentration of 15.25 µg/mL and Trolox 30.5 µg/mL. Therefore, both extracts possess robust antioxidant activity, but IME has been shown to work better than WFE.

#### 2.3.3. DPPH Free Radical Scavenging Activity

Finally, we used the 1,1-diphenyl-2-picrylhydrazyl (DPPH) test to assess the activity of scavenging of extracts. The results obtained were expressed as a percentage of inhibition, and the IC_50_ value represents the concentration of IME and WFE necessary to eliminate 50% of the radical species. After measuring the specific absorbance, the respective IC_50_ values have been calculated. Once again, the extract IME was statistically more effective than WFE: only 3.46 ± 0.061 μg/mL is enough to eliminate 50% of the radicals. On the contrary, it takes 12.45 ± 0.074 μg/mL for WFE to achieve the same result. Figure 6b highlights the results described.

### 2.4. Antioxidant Effect In Vitro

After demonstrating the antioxidant potential of IME and WFE in different assays, we also tested these extracts on an in vitro model to see if it was detectable in cells. Treatment with hydrogen peroxide resulted in a statistically significant accumulation of reactive oxygen species (ROS) compared to untreated cells. IME and WFE treatments did not induce ROS production, demonstrating that they are not pro-oxidants, and their values were similar to those of the untreated cells. Finally, a pre-treatment with IME or WFE, followed by exposure to hydrogen peroxide, demonstrated significant protection against ROS accumulation. In the in vitro experiments, IME also had a greater effect than WFE. These results are shown in Figure 7.

### 2.5. Fibre Contained in WFE and IME

Finally, we studied the fibre content in WFE and IME extracts, evaluating their ligno-cellulosic materials [29]. In particular, we evaluated the following classes of compounds (expressed as a percentage): (1) glucans, (2) xylans, (3) insoluble lignin, (4) soluble lignin, (5) ash, (6) pectin, and other compounds. As can be seen in Figure 8, the percentages of the different compounds in the two extracts are very different. IME presents a compositional profile typical of a plant matrix. The glucan content is very high (55.10%), showing a structural cellulosic predominance. There is also a significant amount of pectin (28.64%), indicating a highly mature cell wall. The xylans (5.19%) and the two insoluble (4.78%) and soluble (3.83%) lignin fractions are present in moderate amounts. Ash amounts to 2.40%. Altogether, the sum of the analysed components reaches a value of about 100%, suggesting that the entire mass is explained by these fractions.

WFE, on the other hand, shows a radically different profile. Glucans are low (9.13%), xylans are undetected, and pectin is almost absent (0.63%). However, the presence of lignin stands out, both in the acid-insoluble (24.95%) and in the soluble form (22.13%). The ash is modest (0.71%). However, the overall mass balance in WFE stands at approximately 57%, thus leaving a missing share of approximately 43%, which by exclusion can be represented by organic acids, proteins, lipids, waxes, sterols, aromatic and phenolic compounds, flavonoids, anthocyanins, and tannins.

## 3. Discussion and Conclusions

This manuscript compares the effects of two extracts (WFE and IME) obtained from the pomegranate fruit and highlights how IME has a higher antioxidant potential than WFE. As has already been discussed, pomegranate has different protective properties for human health thanks to its rich composition of bioactive molecules [30,31]: for example, the numerous polyphenols are responsible for the antioxidant effects [32]. Since there are multiple polyphenols in pomegranate and their presence is assured, we used the method of polyphenol and flavonoid detection (the Folin–Ciocalteu colorimetric assay and the aluminium chloride assay, respectively) to identify the best solvent in which WFE and IME extracts dissolve, as reported in Figure 2 and Figure 3. The characteristics of the extracts may vary depending on the solvent in which they dissolve [33]. Sometimes, it may happen that an extract does not dissolve completely in a solvent, and some micro-components remain intact and absent from the final composition. This may influence the bioavailability of the active compounds and their effectiveness [34]. For this reason, extraction protocols should be validated and standardised [35]. This phenomenon happened in this experimental work: WFE and IME dissolve in different solvents, even though IME is a sub-part of WFE. We expected IME to also dissolve into DMSO, like WFE. On the contrary, the DMSO solvent formed very small but visible lumps in the IME extract, which prevented the extract from reaching the amber colour. The latter was achieved only following the use of a hydro-alcoholic solution (methanol–water, 50:50) together with the disappearance of the visible lumps. Similarly, only DMSO was able to dissolve the lumps present in the WFE extract and reach the purple colour typical of pomegranate pulp [36]. On the other hand, the effects resulting from different solvents on an extract derived from plants have already been treated by our research group [37].

The effects of WFE and IME on cell viability appear free of harmful activity, and the values are comparable (Figure 4). These results suggest that the chemical composition of the extracts may not be dissimilar, also because IME is completely contained in WFE. These results are in agreement with the scientific literature, which reports multiple protective effects of pomegranate extracts on neurons and oligodendrocytes. In fact, the extracts and many metabolites of the fruit support redox balance and regulate proliferation, survival, and cell signalling [38]. Neuroprotective effects are mediated by their antioxidant, anti-inflammatory, and chelating properties, capable of regulating autophagy, apoptosis, and neurotransmitter signalling [39,40]. These protective effects also occur indirectly, and pomegranate extracts can influence the function of the blood–brain barrier by restoring redox balance in the blood and brain and increasing blood flow to the brain [41].

On the contrary, the antioxidant effects of WFE and IME appear different, but with the same significant trend reported throughout the manuscript: IME is always more effective than WFE (Figure 5, Figure 6 and Figure 7). While the antioxidant effects of WFE are widely and commonly reported in the literature [42,43], very little is known about IME, which has a respectable polyphenolic content [44], but has never been tested and compared with another extract from pomegranate.

The greater antioxidant effect of IME cannot be explained by the amount of polyphenols and flavonoids in the extract, which are lower than those of WFE ones (polyphenols: 68 mg GAE/g for WFE; 47 mg GAE/g for IME; flavonoids: 51 mg QE/g for WFE; 35 mg QE/g for IME). Since it has been demonstrated that dietary fibre can exert an antioxidant effect [29,45,46], we have studied this component in both WFE and IME. As can be appreciated in Figure 8, IME is significantly richer in certain fibrous components, such as glucans, xylans, and pectin, than WFE. These fibre fractions are complex carbohydrates indigestible to humans: they are not broken down by digestive enzymes in the upper gastrointestinal tract but reach the large intestine largely intact. [47]. Once in this intestinal portion, they are fermented by the intestinal bacterial flora, acting as prebiotics that promote the growth of beneficial bacteria in the intestine, limit the development of dangerous bacteria, and produce bioactive metabolites such as short-chain fatty acids (SCFAs) [48].

However, the antioxidant effect was appreciated even without involving the intestinal microbiota [49,50,51]; therefore, the robust amount of fibre contained in IME, compared to WFE, could justify its greater antioxidant effect. Although WFE is characterised by a smaller quantity of fibre, it is made up of approximately 43% of “other components,” presumably represented by organic acids, proteins, lipids, waxes, sterols, phenolic compounds, flavonoids, anthocyanins, and tannins. These components may be responsible not only for the antioxidant activity demonstrated, but also for the other protective effects performed by WFE. This study offers two important considerations to the scientific community interested in this topic:(1)Pomegranate must be considered an exceptional fruit capable of performing countless beneficial properties on human health, thanks to a multitude of bioactive compounds [52,53]. The internal membranes, an inedible portion and waste material, represent a precious source rich in fibres, which is essential not only to prevent intestinal diseases, control body weight, regulate blood sugar and cholesterol, but also to activate the Nrf2 signalling pathway or generate a prebiotic effect on the gut microbiota, which plays a beneficial antioxidant effect [54,55].(2)When the gut microbiota is altered, a condition known as “dysbiosis”, a pathological condition, occurs; it affects not only the digestive system, but also the entire organism, generating cardiovascular, neurological, oncological, and psychiatric dysfunctions [56]. Preclinical and clinical studies have confirmed that intestinal dysbiosis is responsible for the pathogenesis of central nervous system disorders [57]. The gastrointestinal tract and the central nervous system can interact via the gut–brain axis, which ensures two-way communication [58]. There are already numerous works that state that dietary fibre can help balance the nervous system, preventing neurodegeneration [59], and that a balanced microbiota is associated with better brain function and a lower risk of neuroinflammation [60]. Since fibre consumption has beneficial effects on the gut microbiota and hinders intestinal dysbiosis, it is reasonable to conclude that fibre can help balance the nervous system and prevent neurodegeneration. Therefore, with these premises, we can hypothesise that the consumption of the internal membranes of the pomegranate can also expand its benefits to the nervous system.

The continuation of this work will involve both in vitro and in vivo studies with models that mimic Alzheimer’s disease to validate the effects resulting from the administration of IME in these experimental conditions. 

In conclusion, if the results confirm the protective role of IME, it would be desirable to create a formulation that facilitates the intake of the internal membranes of the pomegranate, so as to be able to undertake clinical trials.

Finally, it is important to remember that although there are numerous data on the protective effects of pomegranate extracts, there are very few results on the extract from the internal membranes of the fruit [44]. In this manuscript, these two extracts are compared for the first time.

## 4. Materials and Methods

### 4.1. Plant Material and Sample Preparation

The pomegranate fruit (*P. granatum*) was harvested in Soverato, a small town in the province of Catanzaro, Calabria, Italy, at a latitude of 34°36′48′′ E. The harvest was carried out in October 2024 (temperature of 15 °C). The taxonomic identification was confirmed by Professor Salvatore Ragusa, University “Magna Grecia” of Catanzaro (author of this manuscript). The voucher specimen was deposited in the Department of Health Sciences, University “Magna Graecia” of Catanzaro under the following accession number: *Punica granatum* L.13. First, the pomegranates were washed with Milli-Q water to remove pollution or other contaminants. The fruit was then fully squeezed in water and centrifuged (5000 rpm, 10′). The supernatant was filtered through a PVDF syringe filter (pore size 0.45 µm), while the pellet was treated again as described. By this method, the hydrophilic portions have been dissolved, while the larger or slightly hydrophilic content has appeared as tiny, suspended fractions. The extract obtained was subsequently dried to obtain a powder, and 10 mg was dissolved in 1 mL of DMSO. Although the extract was obtained in water, we ensured the dissolution of the micro-particles that had remained intact. The extract has been identified as WFE.

The internal membranes of the pomegranate were gently separated from the peel and dried in an oven (50 °C, for 3 days). The dried samples were ground in a coffee grinder and stored at −80 °C until used. 10 mg was weighed and extracted three times with 1 mL of methanol–water (50:50 *v*/*v*) by sonication (10 min), followed by centrifugation (5000 rpm) for 10 min. The supernatants were pooled together and filtered through a PVDF syringe filter (pore size 0.45 µm). The filtered supernatant extract was used for analysis.

### 4.2. Cell Cultures

For the experiments conducted on a cell line, we used human neurons SH-SY5Y. We purchased these cells from the American Type Culture Collection and cultured them in Eagle’s minimum essential medium supplemented with 10% of bovine foetal serum, non-essential amino acids, penicillin (100 IU/mL), and streptomycin (100 μg/mL). To ensure a cellular phenotype similar to the physiological conditions, we differentiated cells using 10 µM of all-trans-retinoic acid for 5 days. When cells reached 60% confluence, they were treated with WFE or IME for 24 h. Alternatively, the cells were pre-treated with WBF or IME for 24 h and then exposed to H_2_O_2_ (200 µM, for 20 min).

### 4.3. Cell Viability Measurement

Cell viability was measured by the colourimetric test 3-(4,5-dimethylthiazol-2-yl)-2,5-diphenyltetrazolium bromide (MTT). Cells (8 × 10^3^) were seeded into each well of a 96-well plate and treated as described. After 24 h, the medium was replaced with phenol-free medium containing a solution of MTT (0.5 mg/mL), and after 4h of incubation, 100 μL of 10% SDS was added to solubilise the formazan crystals. Optical density was measured through a spectrophotometric reader (X MARK Microplate Bio-Rad, Hercules, CA, USA) at 540 and 690 nm wavelengths.

### 4.4. Reducing Power Assay

The reducing power of WBF and IME extracts was measured by a spectrophotometric test in which the ability to transform Fe^3+^ into Fe^2+^ was measured using the previously reported method [61]. Ascorbic acid was used as a standard: several concentrations of the standard (0.01–0.32 mg/mL) and our extracts (100 μg/mL) were added to 1.0 mL of deionised water. Subsequently, 2.5 mL of phosphate buffer at pH 6.6 and 2.5 mL of potassium ferricyanide (1%) were added. These solutions were incubated at 50 °C for 20 min. At the end of the required time, 2.5 mL of trichloroacetic acid (10%) was added, and the solutions were thoroughly mixed and centrifuged at 3000 rpm for 10 min. The supernatant of each sample was mixed with 2.5 mL of distilled water and 0.5 mL of a 0.1% ferric chloride solution freshly prepared. The absorbance of the mixture was read at 700 nm. An increase in the absorbance of the reaction mixture indicated an increase in the reducing power.

### 4.5. Ferrous Ion (Fe^2+)^ Chelating Activity Assay

Another method for assessing the antioxidant potential of a compound is to measure the formation of the Fe^2+^-ferrozine complex, as described above [62]. This reaction was tested in the concentration range 0.0625–2 mg/mL, using ascorbic acid as a reference standard. 1 mL of each sample was mixed with 1 mL of methanol, 0.1 mL of 2 mM FeCl_2_, and 0.2 mL of 5 mM ferrozine. The resulting solutions were kept in the dark at room temperature for 10 min, and the absorbance was measured at 562 nm. Results obtained are reported as inhibition of the formation of ferrozine-Fe^2+^ complex (%).

### 4.6. ORAC Assay

The antioxidant capacity of WBF and IME is evaluated by measuring the transfer of hydrogen atoms and by measuring the fluorescence loss over time of the fluorescein (used as a probe). The fluorescence reaction is generated spontaneously by the degradation of 2,2-azobis-2-methylpropanimidamide dihydrochloride (AAPH), which occurs at 37 °C. The peroxyl radical oxidises the fluorescein and causes a gradual loss of the fluorescent signal. 6-Hydroxy-2,5,7,8-tetramethylchroman-2-carboxylic acid (trolox) inhibits fluorescence decay. A trolox dose–response curve (7.65, 15.25, 30.5, and 61 μg/mL) was constructed while extracts were used at a concentration of 100 μg/mL. Fluorescein fluorescence decay was measured using a microplate reader at wavelengths of 485 and 520 nm for excitation and emission, respectively. The data obtained showed the antioxidant efficacy of the samples. A regression equation was constructed by comparing the net area under the fluorescein decay curve and Trolox concentration. The area under the curve was calculated with the following equation:i = 90AUC = 1 + Σf1/foi = 1

### 4.7. Determination of Total Phenolic and Flavonoid Content

The total content of polyphenols and flavonoids in WBF and IME extracts was calculated using the Folin–Ciocalteu colorimetric assay and the aluminium chloride assay, respectively. For the quantification of polyphenols, we used several solutions of gallic acid. 400 μL of WBF and IME were mixed with 0.8 mL of 10 times diluted Folin–Ciocalteu reagent. The samples were stirred for 3′ and 0.8 mL of sodium carbonate 7% (*w*/*v*) was added. The resulting solution was left to stand for another 2 h under constant stirring. At the end of the pre-set time, the absorbance was measured at 760 nm, and the total phenolic content of the extracts was expressed in mg equivalent gallic acid (GAE)/g dry weight. The content of flavonoids was determined by the colorimetric test of aluminium chloride. Experimentally, 1 mL of extract was mixed with 1 mL of 2% aluminium chloride in methanol. After 30 min, the absorbance was read at 430 nm, and results were expressed in mg equivalent quercetin (mg QE/g extract)/g dry weight.

### 4.8. ROS Accumulation Measurement

A fluorescent molecule used for the measurement of cellular ROS was fluorescein (H_2_DCF-DA); this compound penetrates inside the cell and is cleaved to form H_2_DCF from intracellular esterases. H_2_DCF remains trapped in the cell and binds to ROS, turning into the highly fluorescent molecule DCF. The quantification of DCF is proportional to the content of ROS. Experimentally, 6 × 10^4^ cells/well were seeded in 96-well microplates. After 24 h, the cells were treated with extracts, and the next day, the growth medium was replaced with fresh medium containing H_2_DCF-DA (25 μM) for 30 min. The cells were exposed or not to H_2_O_2_ (200 μM, 20 min). Finally, the cells were subjected to a spectrofluorimetric analysis (X MARK Microplate Bio-Rad, Hercules, CA, USA).

### 4.9. Fibre Analysis

The analysis of lignocellulosic materials for sugars, lignin, and ash content was performed using the NREL protocol described by Sluiter et al. [63]. The carbohydrate content was determined through a two-step acid hydrolysis method. Initially, the sample was treated with 72% sulfuric acid at 30 °C for 1 h to break down polysaccharide chains into oligomers and monomers. This was followed by a second step involving 3% sulfuric acid at 121 °C for 1 h in an autoclave, ensuring the complete conversion of oligomers to monomers. These monomers were then analysed using ion chromatography (HPIC) with a DIONEX DX300 chromatograph, a Nucleogel Ion 300 OA column, a refractive index ED50 detector, and 0.05 M H_2_SO_4_ as the mobile phase (40 °C, 0.4 mL/min). The acid-insoluble lignin content was measured gravimetrically by filtering the residue with Whatman GFA filters. Acid-soluble lignin was quantified using a Varian Cary 500 spectrophotometer at a wavelength of 205 nm. Ash content was assessed by placing the sample in a muffle furnace at 575 °C overnight. Each characterisation process was carried out in duplicate and represents the average value with the standard deviation.

### 4.10. Statistical Analysis

Data are the mean ± SD and were analysed using GraphPad Prism version 10 (GraphPad Software, San Diego, CA, USA). Comparisons between two groups were performed using the unpaired Student’s *t*-test, while differences among multiple groups were assessed using analysis of Variance ANOVA followed by a Tukey–Kramer comparison test. The number of independent experiments is specified in the figure legends. One, two, and three symbols refer to statistical probabilities (*p*) and indicate *p* < 0.05, *p* < 0.01, and *p* < 0.001, respectively. A *p*-value < 0.05 was considered statistically significant.

## Figures and Tables

**Figure 1 ijms-26-09807-f001:**
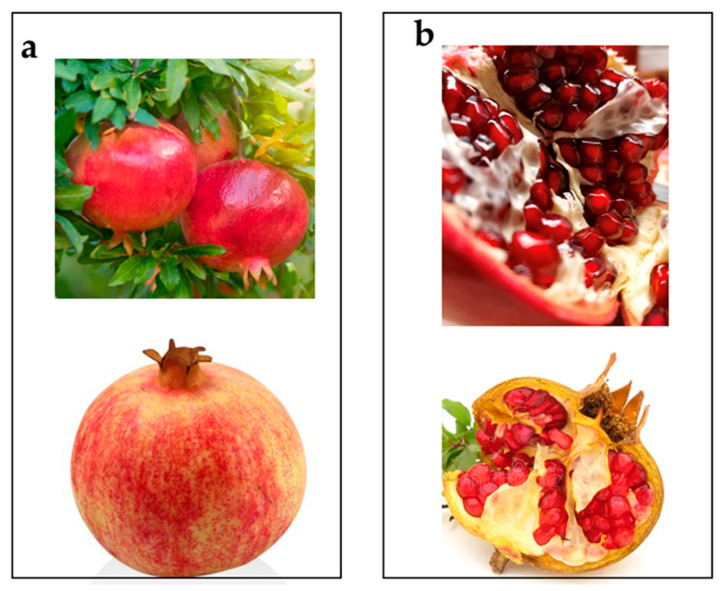
Pomegranate parts and extracts. Two images of the whole pomegranate fruit are represented in panel (**a**). WFE extract was obtained from the entire fruit, including all districts. The internal membranes separating the pomegranate lodges were represented in panel (**b**), and IME comes from their extraction.

**Figure 2 ijms-26-09807-f002:**
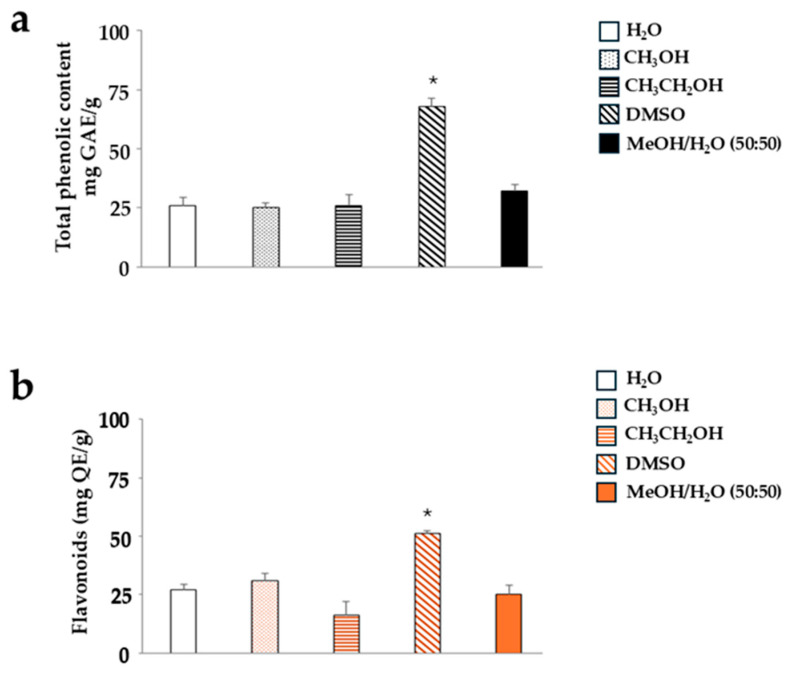
Yield of polyphenols and flavonoids in several solvents. Panel (**a**) shows the content of polyphenols, while panel (**b**) shows the amount of flavonoids. In both cases, the extract of interest is WFE. Three independent experiments were performed, and the values are expressed as the mean ± SD. * denotes *p* < 0.05 vs. the extract dissolved in water. The statistics were carried out using the *t*-test.

**Figure 3 ijms-26-09807-f003:**
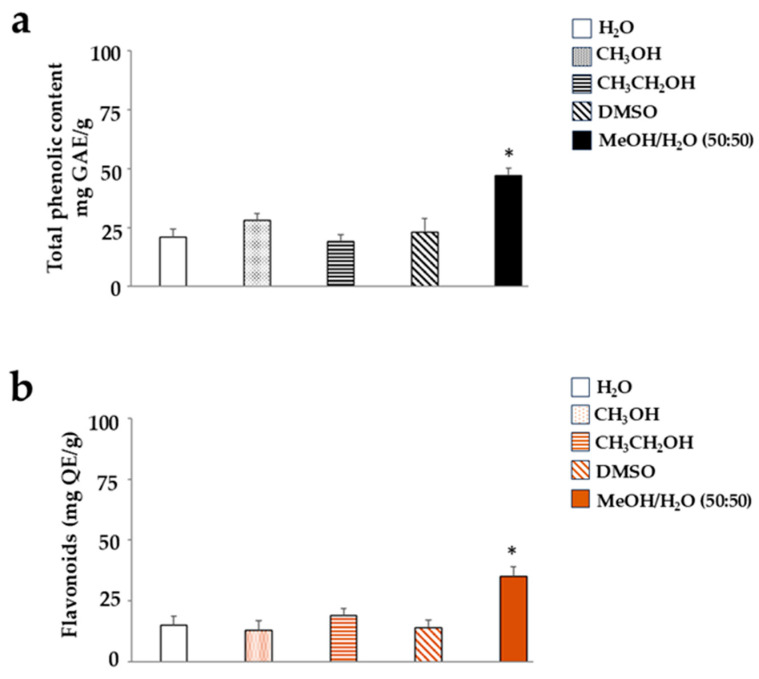
Yield of polyphenols and flavonoids in IME extract. Panel (**a**) shows the content of polyphenols, while panel (**b**) shows the amount of flavonoids. In both cases, the solvents indicated are used. Three independent experiments were performed, and the values are expressed as the mean ± SD. * Denotes *p* < 0.05 vs. the extract dissolved in water. The statistics were carried out using the *t*-test.

**Figure 4 ijms-26-09807-f004:**
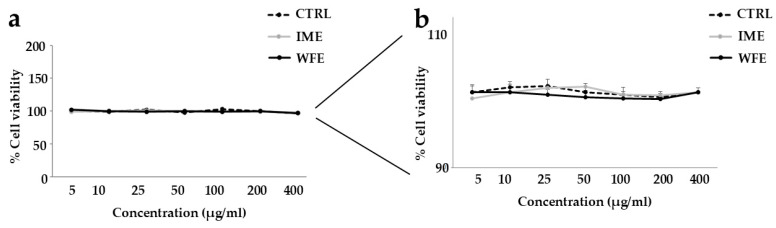
Selection of the concentration to be used for WFE and IME. In Figure 4 panel (**a**), the cell viability of cells treated with different concentrations of extracts was represented. Since the values obtained were superimposed, an appropriate magnification was reported in panel (**b**). The obtained values were compared to those of untreated cells (CTRL).

**Figure 5 ijms-26-09807-f005:**
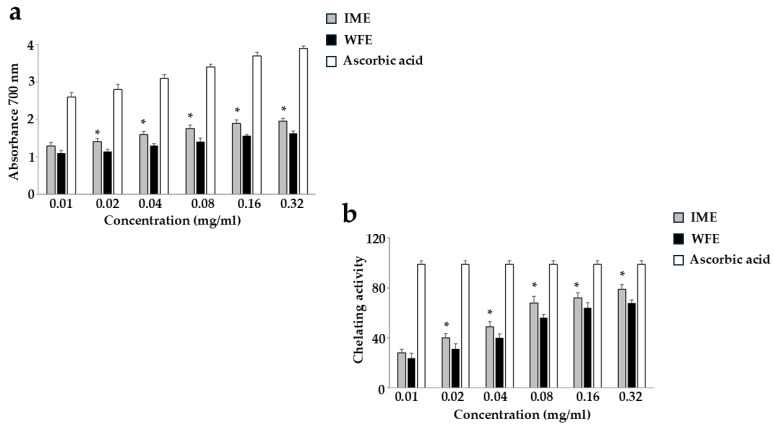
Reducing Power and Ferrous Ion (Fe^2+^) Chelating Activity. In Figure 5, the oxidative potential of IME and WFE is shown. In panel (**a**), the ability of WFE and IME to reduce ferric ions to ferrous ions, donating electrons and neutralising free radicals, was measured. Ascorbic acid, administered at increasing concentrations, is the positive control. Panel (**b**) highlights the ability of extracts to bind to ferrous ions and inhibit their activity in promoting oxidative damage. Three independent experiments were performed, and the values are expressed as the mean ± SD. * Denotes *p* < 0.05 vs. the extract WFE at the same concentration. A Tukey–Kramer comparison test followed the analysis of Variance (ANOVA).

**Figure 6 ijms-26-09807-f006:**
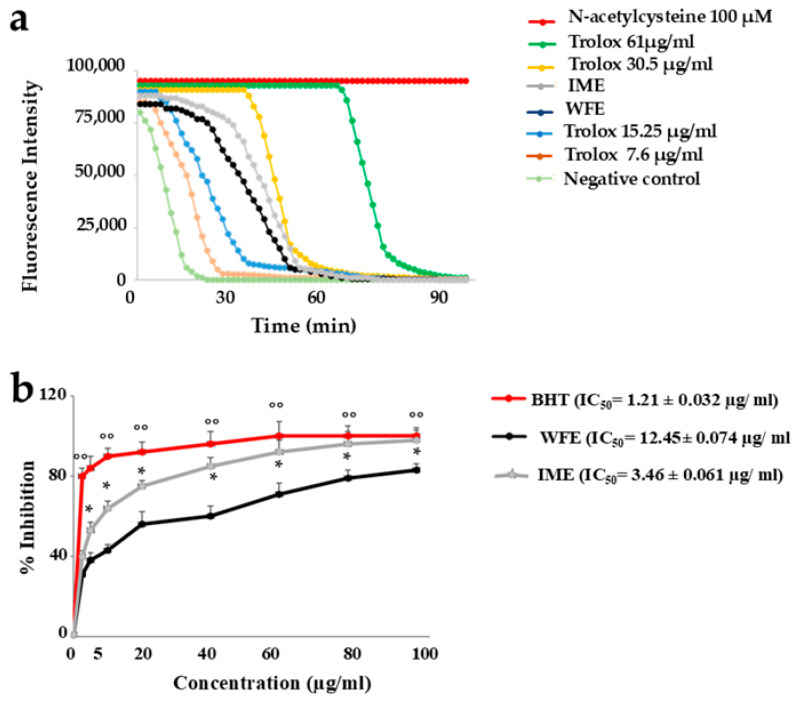
Further measurement of the antioxidant potential of IME and WFE through other assays. Panel (**a**) shows the oxygen radical absorbance capacity of the extracts. Three independent experiments were performed, and the values are expressed as means. Panel (**b**) highlights the concentration of IME and WFE necessary to eliminate 50% of the radical species. Three independent experiments were carried out, and the values are expressed as the mean ± SD. * denotes *p* < 0.05 vs. the extract WFE at the same concentration; °° denotes *p* < 0.01 vs. the extract WFE at the same concentration. A Tukey–Kramer comparison test followed the analysis of Variance (ANOVA).

**Figure 7 ijms-26-09807-f007:**
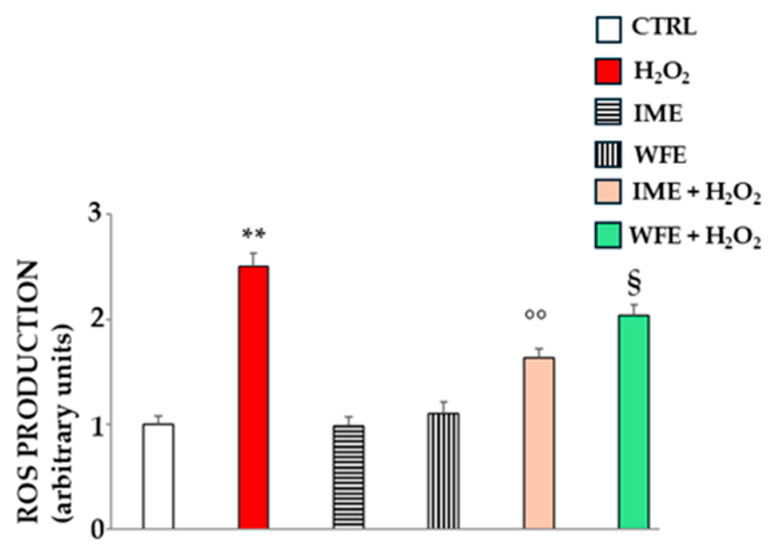
Antioxidant effects of IME and WFE in vitro. The exposure of cells to hydrogen peroxide 200 µM for 20 min leads to a significant increase in ROS. IME and WFE alone are not pro-oxidants, while pre-treatment with extracts followed by exposure to hydrogen peroxide significantly reduces the accumulation of ROS. Three independent experiments were carried out, and the values are expressed as the mean ± SD. ** denotes *p* < 0.01 vs. untreated cells; °° denotes *p* < 0.01 vs. hydrogen peroxide; § Denotes *p* < 0.05 vs. hydrogen peroxide. A Tukey–Kramer comparison test followed the analysis of Variance (ANOVA).

**Figure 8 ijms-26-09807-f008:**
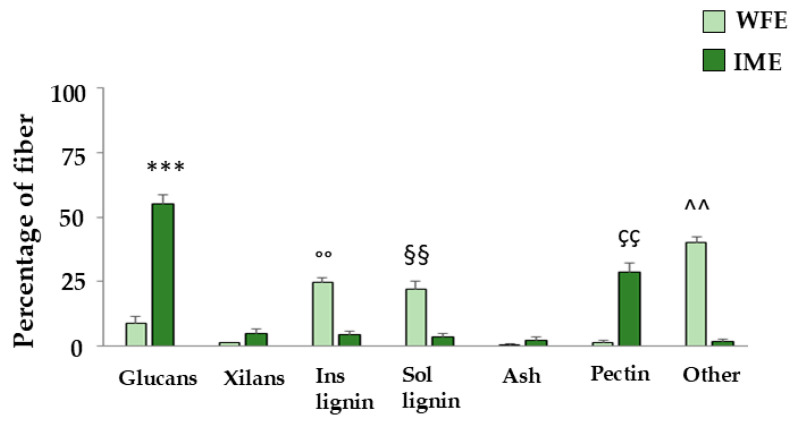
Qualitative and quantitative evaluation of fibre in WFE and IME. Results for some fibre families (glucans, xylans, insoluble and soluble lignin, ash, pectin, and other compounds) are represented. Three independent experiments were performed, and the values were expressed as the mean ± SD. *** denotes *p* < 0.001 vs. Glucans of WFE; °° denotes *p* < 0.01 vs. Ins lignin of IME; §§ denotes *p* < 0.01 vs. Sol Lignin of IME; çç denotes *p* < 0.01 vs. Pectin of WFE; ^^ denotes *p* < 0.01 vs. Other of IME. A Tukey comparison test followed variance analysis (ANOVA).

## Data Availability

The raw data supporting the conclusions of this article will be made available by the authors on request.

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
