# Peer review of "Polyphenols and Fibre: Key Players with Antioxidant Activity in Two Extracts from Pomegranate (Punica granatum)"

_ijms, 2025, doi:10.3390/ijms26199807_

Round 1
Reviewer 1 Report
Comments and Suggestions for Authors
In this paper, the authors compared antioxidant activities in two extracts from pomegranate. This topic is interesting. The research addresses a relevant challenge, and the experimental design is reasonable. Overall, the data collections, the figures and table, the results and discussions were reasonable. The references are appropriate. I would like to recommend it for publication after minor revisions. The detailed comments are as follows:
- Line 23, WFE and IME mean what? Define all abbreviations upon first use.
- Fig. 2a and 2b, Fig. 3a and 3b need to be aligned. The number needs to be subscript.
- In Fig. 2b MeOH/H2O 70:30. But Fig. 2a, 3a, 3b, 50:50. Please check.
- Why these solvents were chosen and how may them differentially affect extract composition?
- No error bars in Fig. 4? Please add.
- Add some discussion on how the measured values compare to other related studies.
- Check reference styles. For example, line 458, no page numbers.
Author Response
Reviewer 1
Dear reviewer, thank you for your valuable suggestions. I have revised the manuscript as requested, and you can see your comments highlighted in yellow and my responses in green.
In this paper, the authors compared antioxidant activities in two extracts from pomegranate. This topic is interesting. The research addresses a relevant challenge, and the experimental design is reasonable. Overall, the data collections, the figures and table, the results and discussions were reasonable. The references are appropriate. I would like to recommend it for publication after minor revisions. The detailed comments are as follows:
- Line 23, WFE and IME mean what? Define all abbreviations upon first use.
Abbreviations of extracts have been defined (Line 23)
- Fig. 2a and 2b, Fig. 3a and 3b need to be aligned. The number needs to be subscript.
Figures have been modified as required.
- In Fig. 2b MeOH/H2O 70:30. But Fig. 2a, 3a, 3b, 50:50. Please check.
The methanol-water ratio (50:50) was checked and corrected.
- Why these solvents were chosen and how may them differentially affect extract composition?
Greater clarification has been added in the manuscript (Lines 117-124). These solvents could influence the amount of polyphenols and flavonoids extracted in relation to their ability to bind molecules with different solubilities.
- No error bars in Fig. 4? Please add.
The correct figure 4 has been inserted.
- Add some discussion on how the measured values compare to other related studies.
A comparison between the results obtained and the data already published has been reported. (Lines 293-305).
- Check reference styles. For example, line 458, no page numbers.
Reference styles were checked and corrected. (Line 511).
Reviewer 2 Report
Comments and Suggestions for Authors
The manuscript entitled “Polyphenols and fiber: key players with antioxidant activity in two extracts from pomegranate (Punica granatum)” is clearly written and presents interesting findings. However, a few clarifications and improvements are required before it can be considered for publication:
In figure 4, please specify what was used as the control in this experiment. Since WFE and IME were dissolved in different solvents, it is important to clarify whether separate solvent controls were used or whether a single control was applied. This will help readers correctly interpret the results.
In section 2.3, expand on this and the corresponding results should be explicitly referenced. This will strengthen the connection between methods and findings.
Author Response
Reviewer 2
Dear reviewer, thank you for your valuable suggestions. I have revised the manuscript as requested, and you can see your comments highlighted in yellow and my responses in green.
The manuscript entitled “Polyphenols and fiber: key players with antioxidant activity in two extracts from pomegranate (Punica granatum)” is clearly written and presents interesting findings. However, a few clarifications and improvements are required before it can be considered for publication:
In figure 4, please specify what was used as the control in this experiment.
The obtained values were compared to untreated cells (CTRL) and this sentence was added in the caption of figure 4 (Lines 163-164).
Since WFE and IME were dissolved in different solvents, it is important to clarify whether separate solvent controls were used or whether a single control was applied. This will help readers correctly interpret the results.
Control cells were exposed to vehicles used to dissolve WFE and IME and did not affect the results (data not shown). Thus, the control depicted in the figures consists of untreated cells. This clarification was included in the main text (Lines 144-146).
In section 2.3, expand on this and the corresponding results should be explicitly referenced. This will strengthen the connection between methods and findings.
Section 2.3 has been expanded and referenced, as required.
Reviewer 3 Report
Comments and Suggestions for Authors
The article examines the antioxidant activity of two pomegranate extracts (WFE and IME), reporting higher activity in IME despite its lower polyphenol and flavonoid content, which the authors attribute to differences in fiber composition.
While the methodology is clearly described, including the statistical analysis, and the data are properly treated, the manuscript lacks direct comparison with previously published studies on pomegranate. The Discussion section would be strengthened by contextualizing the findings within the existing literature, indicating whether similar results have been reported, how this study differs from them, and whether the results are consistent with what is already known for this plant.The authors should consider adding references and discussing their results in relation to other studies previously published on pomegranate
In the methodology section, the procedure of the statistical analysis should also be described.
Author Response
Reviewer 3
Dear reviewer, thank you for your valuable suggestions. I have revised the manuscript as requested, and you can see your comments highlighted in yellow and my responses in green.
The article examines the antioxidant activity of two pomegranate extracts (WFE and IME), reporting higher activity in IME despite its lower polyphenol and flavonoid content, which the authors attribute to differences in fiber composition.
While the methodology is clearly described, including the statistical analysis, and the data are properly treated, the manuscript lacks direct comparison with previously published studies on pomegranate. The Discussion section would be strengthened by contextualizing the findings within the existing literature, indicating whether similar results have been reported, how this study differs from them, and whether the results are consistent with what is already known for this plant.The authors should consider adding references and discussing their results in relation to other studies previously published on pomegranate.
A comparison between the results obtained and the data already published has been reported. The discussion section and references have been enriched (Lines 291-305).
In the methodology section, the procedure of the statistical analysis should also be described.
Statistical analysis was described as required in Section 4.10, Lines 483-489
The English language was checked and improved by a native speaker contributor.